# The Enhanced Energy Density of rGO/TiO$_2$ Based Nanocomposite as Electrode Material for Supercapacitor

**Palani Anandhi** [1], **Santhanam Harikrishnan** [2], **Veerabadran Jawahar Senthil Kumar** [3], **Wen-Cheng Lai** [4,5,*] and **Alaa El Din Mahmoud** [6,7]

[1] Department of Electronics and Communication Engineering, Saveetha School of Engineering, Saveetha Institute of Medical and Technical Sciences, Chennai 602105, India; anandhi_electro@yahoo.com
[2] Department of Mechanical Engineering, Kings Engineering College, Chennai 602117, India; ramhkn@yahoo.co.in
[3] Department of Electronics and Communication Engineering, Anna University, Chennai 600025, India; veerajawahar@gmail.com
[4] Bachelor Program in Industrial Projects, National Yunlin University of Science and Technology, Douliu 640301, Taiwan
[5] Department Electronic Engineering, National Yunlin University of Science and Technology, Douliu 640301, Taiwan
[6] Environmental Sciences Department, Faculty of Science, Alexandria University, Alexandria 21511, Egypt; alaa-mahmoud@alexu.edu.eg
[7] Green Technology Group, Faculty of Science, Alexandria University, Alexandria 21511, Egypt
* Correspondence: wenlai@yuntech.edu.tw or wenlai@mail.ntust.edu.tw

**Abstract:** TiO$_2$ electrode material is a poor choice for supercapacitor electrodes because it has low conductivity, poor cyclic stability, and a low capacitance value. It is inevitable to enhance electrode materials of this kind by increasing the surface area and combining high electronic conductivity materials. In the current research work, it was proposed to combine reduced graphene oxide (rGO) as it might provide a large surface area for intercalation and deintercalation, and also, it could establish the shorter paths to ion transfer, leading to a reduction in ionic resistance. The size, surface morphology, and crystalline structure of as-prepared rGO/TiO$_2$ nanocomposites were studied using HRTEM, FESEM, and XRD, respectively. Using an electrochemical workstation, the capacitive behaviors of the rGO/TiO$_2$ electrode materials were assessed with respect to scan rate and current density. The capacitances obtained through cyclic voltammetry and galvanostatic charge-discharge techniques were found to be higher when compared to TiO$_2$ alone. Furthermore, the as-synthesized nanocomposites were able to achieve a higher energy density and better cycle stability.

**Keywords:** nanocomposite; electrode material; supercapacitor

## 1. Introduction

The consumption and generation of energy produced by the combustion of fossil fuels are expected to increase in the next decades. The carbon dioxide emissions from fossil fuel burning are anticipated to have a significant influence on global economics and the environment. However, non-carbon-emitting sources, such as solar, wind and nuclear generate electrical power for hybrid as well as all-electric vehicles, offering the potential of providing a solution to the current energy crisis. However, efficient electrical energy storage is required to utilize the electricity generated from these renewable sources. Furthermore, considerably improved electrical energy storage systems are required to supply the continuous energy demand while also efficiently leveling the cyclic nature of various energy sources. Additionally, reliability and safety are required to avoid premature pavement. Electrochemical energy generation is being considered as an alternative energy source, provided that it is more sustainable and eco-friendlier [1].

Fuel cells, batteries, and supercapacitors are all electrochemical energy storage systems. Even though the energy storage and conversion methods of these three systems are distinct, they all have "electrochemical similarities". It is noted that supercapacitors offer high power density, high energy storage, and extended cycle life [2,3]. It has higher capacitance values than capacitors and stores a large energy per unit volume than electrolyte capacitors. As well as having faster charge storage and delivery, they also have a longer lifespan than rechargeable batteries [4]. However, the specific energy of supercapacitors is less than batteries and fuel cells. For any application, they demand a supply of energy for long durations, which requires the use of batteries or other power sources in combination with supercapacitors [5]. Therefore, there is a significant focus on enhancing the energy density of supercapacitors to a rate that is closer to that of batteries. There are two basic forms of supercapacitors, which are characterized by the methods of energy storage that utilize electric double-layer capacitors (EDLCs) and pseudo capacitors. EDLCs could store charge in two ways: electrostatically or by a non-faradic technique that does not need the transfer of charge between both the electrode and the electrolyte. Generally, carbon materials or derivatives, such as graphene, CNT, etc., are used as electrode materials for EDLCs and these materials provide a high surface area and long cycling stability. On the other hand, the specific energy in the EDLC is poor due to the electrostatic surface charging process that takes place [6]. Recently, researchers have focused on the progress of new materials that have the potential to enhance the energy performance of EDLC. Pseudo capacitors store energy by faradaic reaction, intercalation, or electrosorption in which the charge transfer between the electrode-electrolyte takes place due to a faradic current flowing through the supercapacitor. Moreover, it has a higher specific capacitance and specific energy than EDLC due to a faradic reaction. The most commonly used materials for pseudo capacitors are transition metal oxides ($MnO_2$, $RuO_2$, $Co_3O_4$, NiO, ZnO, $TiO_2$, $V_2O_5$, and transition metal sulfides) and conducting polymers [7–10]. Conducting polymers have significant features, such as low cost, low environmental effect, high conductivity, a high operating voltage window, a large amount of storage capacity, high porosity, and reversibility. The specific capacitance of polymer materials is achieved by a redox reaction. Unlike carbon electrodes, charging/discharging in conducting polymers involves the entire material. During oxidation, ions are transported to the polymer structure; during reduction, ions are released into the electrolyte. The charging and discharging reactions are extremely reversible and there is no change in the structure, such as phase changes during the redox reaction. However, during charging and discharging, the ion diffusion in the polymer changes its volume, imposing enormous stress on the polymer's structure, due to this polymer can swell and shrink, leading to mechanical disintegration. As the cycle number increases, the polymer structure collapses finally and as a result, the capacitance of the polymer is reduced. So, the polymer materials have relatively poor cycling stability and a low capacitance value when compared to metal oxides, however, they provide higher energy density than metal oxides [11,12]. Therefore, for enhancing the electrochemical performance of the polymer materials, they are combined with carbon and metal oxide materials. Metal oxides are used in supercapacitors because of their high conductivity, large specific capacitance, and long life. When a polymer material is combined with a metal oxide, the composite shows enhanced specific capacitance and greater cyclic stability than pure polymer electrodes. If carbon-based materials are combined with a metal oxide material, it exhibits higher energy density, higher capacitance value, and longer cyclic stability than individual materials.

Graphene derivatives are employed as an electrode material because of the large specific surface areas, high electrical conductivity and the large-scale production of these materials is easier. Graphene oxide is referred to as the oxidized form of graphene. Due to the charged structure and hydrophilicity, GO sheets disperse rapidly in stable aqueous and organic solutions [13]. Alteration of the $sp^2$ bonding network causes the development of various characteristics, such as low conductivity, and it exhibits either an insulating or semiconductive characteristic, with a large surface area of roughly 890 $m^2g^{-1}$. To improve

the electronic conductivity, the reduction of GO is suggested to be a promising way to obtain graphene-like properties in a variety of applications [14,15]. rGO is synthesized by green routes, chemical, thermal, or photothermal reduction methods. In chemical oxidation synthesis, rGO contains a minimum number of oxygens, resulting in its structural change. rGO exhibits high electrical conductivity of 6300 S cm$^{-1}$ than GO and it shows a higher surface area [16].

Among many pseudo capacitors materials, $TiO_2$ is considered an electrode material due to its vast surface area, low cost, availability in abundant amounts, environmentally friendliness, multiple oxidation states (IIIV), and high chemical stability with a wide bandgap semiconductor material [17]. On account of its remarkable physicochemical properties, nano-$TiO_2$ with a high surface area has gained a lot of attention for its capacity to give the highest performance for a variety of applications, such as light detectors, solar cells, and gas sensing. When utilized as the electrode in supercapacitor devices, this material has the potential to achieve a high storage capacity [18]. $TiO_2$ is used in the battery-type electrode due to its high charge storage capacity. However, the practical use of this battery material could not be implemented because of its low electronic conductivity and poor cyclic stability. If the number of cycles is increased then, it could cause a loss of active electrochemical sites [7,16]. Further, pure $TiO_2$ material exhibits high conduction resistivity, so its capacitance value is negligible and not suitable for supercapacitor application. The capacitance of the $TiO_2$ electrode of the supercapacitor could achieve the maximum value of 165 F/g only [19,20] and hence, it paves a path to explore further to enhance the capacitance of the $TiO_2$. The present study aimed to augment the electrochemical performance of $TiO_2$ by combining a high electronic conductivity material. Herein, the wet chemical method was chosen to synthesize $rGO/TiO_2$ nanocomposite as electrode material for the supercapacitor. An addition of rGO into $TiO_2$ reduced the resistance of the electrode material and $TiO_2$ nanospheres were decorated between the surfaces of the rGO sheets. This arrangement seemed to be a sandwich and it provided a larger surface area to facilitate the migration of ions between electrolyte and electrode and vice versa. In addition to this, the wet chemical technique has the benefit of being simple, rapid, and cost-effective in the production of nanocomposites for industrial-scale manufacturing, and it does not require the use of any specialist equipment.

## 2. Design of Materials

From SRL India, we purchased reagent grade chemicals, such as graphite flakes, $Ti(OBu)_4$, $KMnO_4$, $NaNO_3$, ethanol, $H_2SO_4$, HCl, hydrogen peroxide and NaOH pellets.

### 2.1. Synthesis of $TiO_2$

The sol-gel technique was used to synthesize $TiO_2$ nanoparticles from $Ti(OBu)_4$ and ethanol (1:4 volume ratio). The solution was stirred for 15 min at 750 rpm, mixing NaOH pellets into the solution until pH 7 was obtained. NaOH can affect particle size and form throughout the reaction. After that, a white $TiO_2$ precipitate was discovered. The samples were then centrifuged and rinsed three times with 2D water. They were dried at 100 °C for 24 h and then ground with an agate mortar.

### 2.2. Synthesis of GO

Hummer's method was used to produce GO from graphite flake (GF). GF was added to 45 mL of $H_2SO_4$ in an ice bath with 700 rpm magnetic stirring. With constant stirring, 0.75 g of $NaNO_3$ and 4.5 g of $KMnO_4$ were added. After an hour, the solution was maintained at 35 °C for 2 h. The solution was held at 95 °C with nonstop stirring in a water bath for 30 min; 150 mL of water with 4.5 mL $H_2O_2$ was added to purify the mixture. After filtration with HCl and washing with bi-distilled water, this product was kept in an oven at 100 °C for 8 h and ground with an agate mortar to achieve a fine powder.

### 2.3. Synthesis of rGO/TiO₂ Nanocomposites

500 milligrams of manufactured $TiO_2$ nanospheres were dispersed in 50 mL of 2D water and agitated at 90 °C for 30 min to make $rGO/TiO_2$ composites; 0.1 g of synthesized GO was mixed in 30 mL of double-distilled water under sonication for 15 min, then added to the $TiO_2$ solution and agitated for 1 h at 90 °C. The mixture was then stirred for 4 h at 90 °C with 3 g of urea. Water and ethanol were used to wash the samples many times. The composites were dried in an oven at 100 °C for 12 h before annealing [21]. Afterward, the samples were loaded into a high temperature box furnace and annealed at 600 °C for 2 h.

### 2.4. Characterization

HRTEM was used to ensure the synthesized NPs were 1–100 nm in size. NP size and surface could be determined [Philips CM120 series; Eindhoven, Netherlands]. For HRTEM images, drops of colloid were dispersed over a Cu grid and covered with a carbon sheet. Zeiss LEO 1530 FESEM was recommended to analyze the sample morphology and the EDX Spectrum was collected to determine the sample composition. X-ray diffraction of all samples was analyzed with a diffractometer [Brand: Rigaku, Model: Miniflex-II C, Tokyo, Japan] with X-ray radiation of wavelength 1.54056 Å. In the present research work, an electrochemical workstation [Brand: Biologic, Model: VSP Potentiostat, Seyssinet-Pariset, France] was employed to determine the necessary properties of the supercapacitors.

### 2.5. Electrochemical Measurement

As-fabricated $TiO_2$ and $rGO/TiO_2$ samples were two active materials for supercapacitors. The working electrodes were fabricated as per the previous literature [7]. The mass of the samples in the working electrode was estimated at 4.3 mg. The capacitive performance of the electrode sample was measured using Biologic Instruments [Mumbai, India] with three electrode configurations [8,22] and 1M $Na_2SO_4$ aqueous solution with nitrogen gas.

### 2.6. Discussion

The size and surface morphology of the nanoparticles and nanocomposites could determine the behavior of the electrodes employed in the supercapacitor. This could provide information on the active sites required for the migration of ions between the electrode and electrolyte. This may have a significant effect on the electrical conductivity and porosity [23]. The greater surface areas of the electrode materials are beneficial to offering maximum electroactive sites even at higher current densities [24]. In addition, the size of the active materials used in the electrodes of the supercapacitor plays a key role in the values of the specific capacitance. Nanostructures with a larger surface area could reduce the transport pathways for electrons and ions, resulting in better diffusivity. Therefore, it is expected to study the morphologies of the as-synthesized nanomaterials. As the particle size increases, specific capacitance decreases and vice versa. Hence, it is inevitable to find the size of particles. Therefore, the FESEM and HRTEM images were examined.

The FESEM images of $TiO_2$ nanoparticles shown in Figure 1a,b appear as flake-like structures. Figure 1c,d reveals that nano-$TiO_2$ is spread over the surface of the rGO sheets and this morphology could be beneficial for accelerating the ion transfer between the electrode and electrolyte interface. It is noting worthy that Figure 1e indicated the good combination of nano-$TiO_2$ and rGO sheets with the loading of nano-$TiO_2$ on the surface sheet of rGO [25]. Furthermore, the EDX spectrum of the $rGO/TiO_2$ sample (Figure 1e) indicates the existence of C, O and Ti elements in the prepared nanocomposite of $rGO/TiO_2$. Other than the detected elements, no other contaminants are discovered, indicating the purity of the produced nanocomposite. Additionally, HRTEM in Figure 2a,b illustrates the flake-like structure of $TiO_2$ nanoparticles and the average size of the nanoparticles was 47 nm. The $rGO/TiO_2$ nanocomposite is shown in Figure 2c,d and it is divulged that $TiO_2$ particles are spread and aggregated over the rGO sheets. Furthermore, the rGO sheet edge could be identified in Figure 2b. The smaller sizes of $TiO_2$ nanoparticles and the cumulative layered architecture of graphene sheets favor increasing the nanocomposites'

specific surface area which could be advantageous to promoting the electrochemical activity further. Recently, a similar observation was found by Britto et al. [26], who detected $TiO_2$ nanoparticles agglomerated and disseminated on the graphene nanosheets which are few-layered sheets.

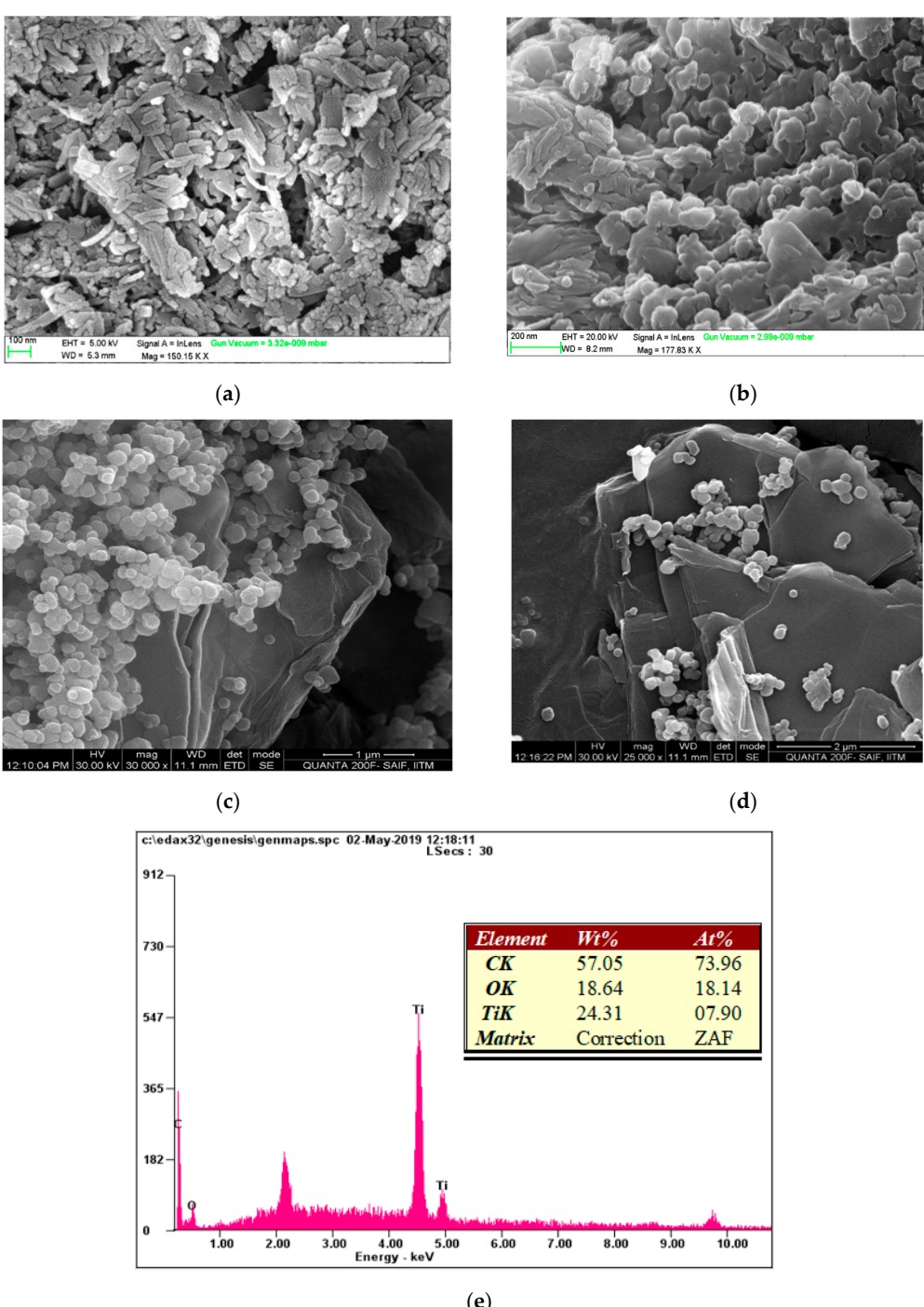

**Figure 1.** (**a**,**b**) FESEM images of $TiO_2$ nanospheres. (**c**,**d**) FESEM images of rGO/$TiO_2$ nanocomposites, (**e**) EDX Spectrum of the rGO/$TiO_2$ nanocomposites with the chemical composition (inset).

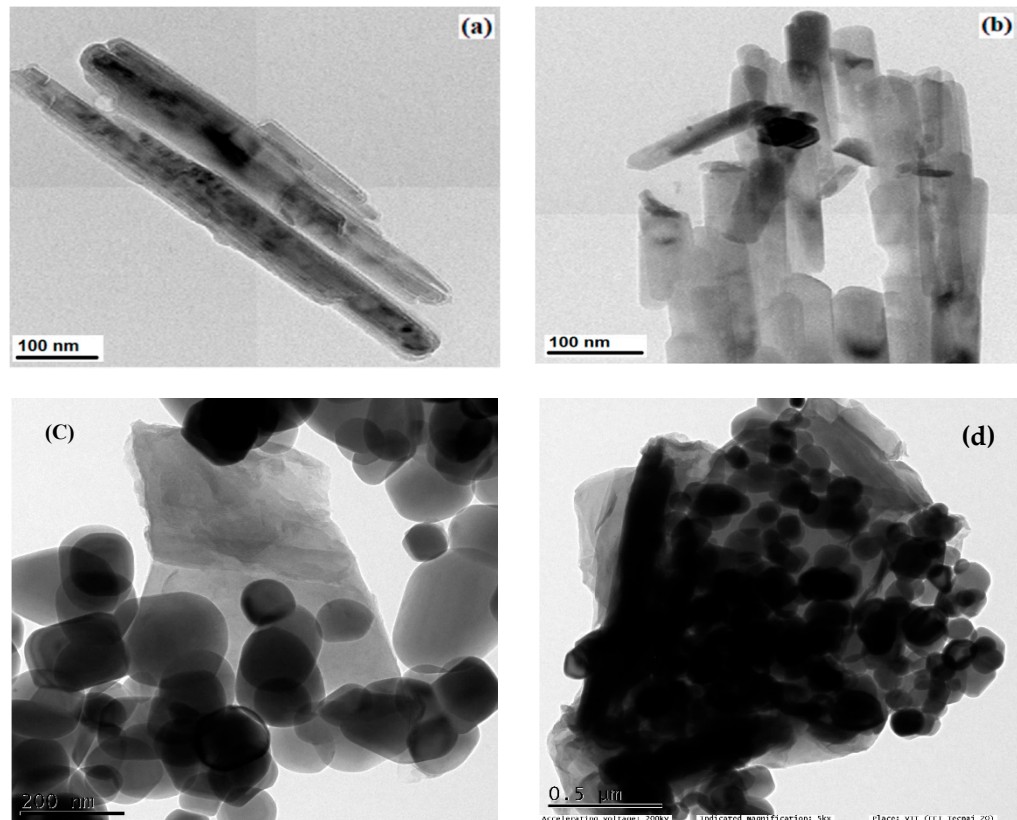

**Figure 2.** (**a**,**b**) HRTEM images of TiO$_2$ nanospheres. (**c**,**d**) HRTEM images of rGO/TiO$_2$ nanocomposites at two different magnifications.

The XRD spectrum discloses the three phases of nano-TiO$_2$ namely, brookite, anatase, and rutile (Figure 3a). The existence of these phases is credited to the annealing temperature at 400 °C. There were no impurity peaks present, which reveals the absence of foreign materials, so the nano-TiO$_2$ is pure and stable.

Figure 3b displays the XRD patterns of synthesized rGO/TiO$_2$ nanocomposites by the wet chemical method with anatase-rutile crystalline phases. The XRD peaks and their intensities for both phases of TiO$_2$ are presented. The absence of the brookite crystalline phase was ascribed to the annealing temperature at 600 °C and at this condition, the brookite phases were converted into anatase phases [27]. The TiO$_2$ with anatase phase showing strong diffraction peaks at 25.3°, 37.7°, 48.23°, 56.63°, and 75.1° are matched to (101), (004), (200), (211), and (215) crystal planes [JCPDS No. 21-1272]. On the other hand, the rutile phase also shows foremost diffraction peaks at 27.4°, 36.1°, 41.4°, 44.2°, 54.0°, 62.7°, 64.1°, 69.2°, and 69.9° corresponding to the crystal planes of (110), (101), (200), (210), (111), (002), (310), (301) and (112) [JCPDS No. 21- 1276], respectively. Because of the weak intensity of rGO, its peak might not be determined by XRD, and hence, no distinctive peaks for rGO were observed. Additionally, the main peaks of rGO at 25° are interfered with or overlapped by a strong (101) peak of the TiO$_2$ anatase phase at 25.3°. These results are matched to those attained by Sundriyal et al. [28] and Olana et al. [25].

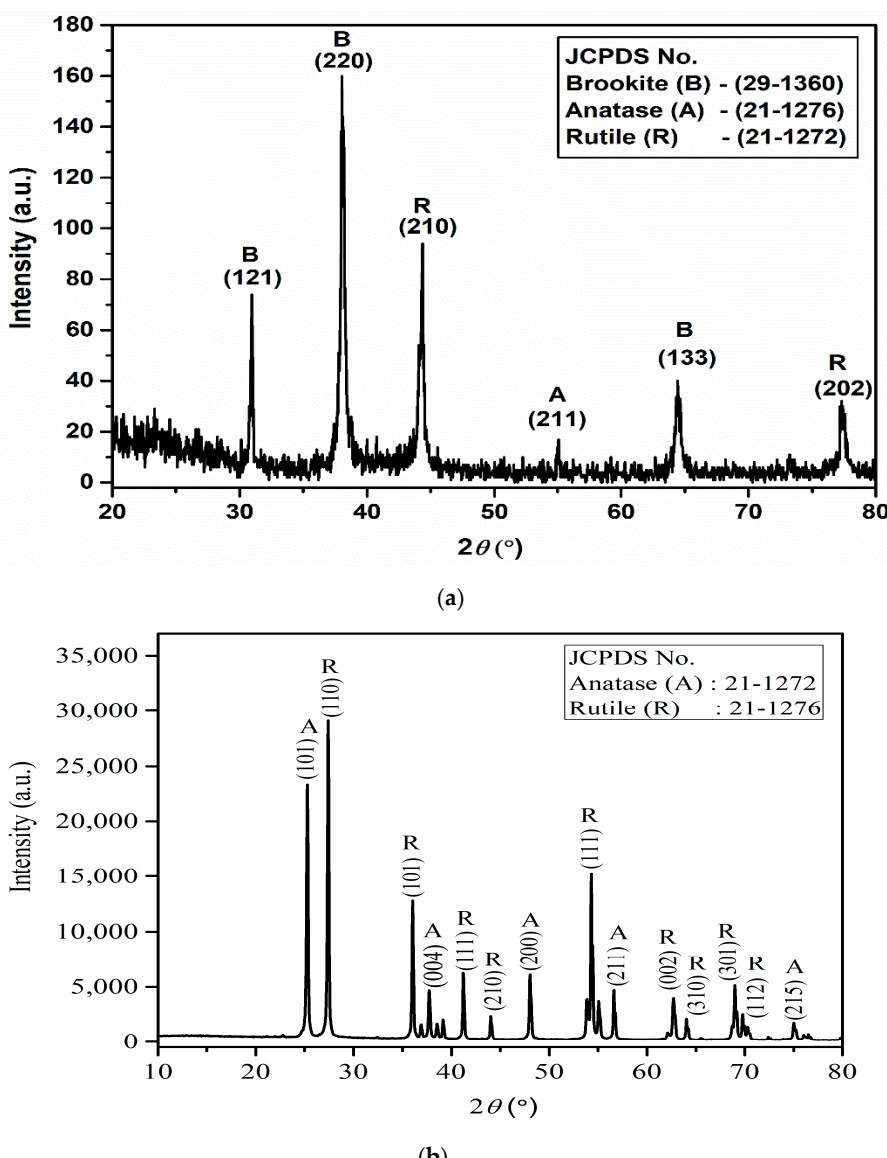

**Figure 3.** XRD spectra of (**a**) $TiO_2$ nanospheres and (**b**) $rGO/TiO_2$ nanocomposites.

### 3. Cyclic Voltammetry (CV)

The electrochemical behavior of $TiO_2$-based electrodes for a supercapacitor was evaluated using CV measurements. Figure 4a,b presents the CV profiles of the $TiO_2$ and $rGO/TiO_2$ nanocomposites, respectively. The CV tests on the samples were conducted at different scan rates varying from 5 to 100 mV s$^{-1}$. The CV profiles of $TiO_2$ differ from the rectangular profile (it is required to confirm the EDLC type) and they show the presence of redox peaks, revealing the pseudocapacitive characteristics of the electrode materials [29]. Further, the capacitance of electrodes is mainly influenced by the faradaic redox reactions. In Figure 4a, the shape of the CV curve of $TiO_2$ shows a slight hump during charging-discharging (oxidation-reduction) and it ascertains the capacitive response of $TiO_2$ resulting from pseudo capacitance. The faradic charge storage initiates the reversible redox reactions at the surface of the electrode by transferring ions with an electrolyte solution. Figure 4b shows the CV profiles of the $rGO/TiO_2$ nanocomposite and the test on the sample was implemented at diverse scan rates ranging from 5 to 100 mV s$^{-1}$. In Figure 4b, it is shown that the CV profiles of $rGO/TiO_2$ appear to be the quasi-rectangular shape which resulted from the combined effect of EDLC and pseudo-capacitance achieved by rGO and $TiO_2$, respectively. In Figure 4b, it is found that there is a noticeable change in the CV profile

of the nanocomposite when compared to the CV profile of $TiO_2$ shown in Figure 4a. The addition of rGO has modified the CV profile of the nanocomposite as it introduces the EDLC behavior and it is observed between 0.5 to 0.8 V. When the scan rate increases, the peaks at the anode and cathode move towards higher and lower voltages, respectively. The phenomenon resulted in improved polarization and kinetics of the electrolyte ions' access to the electrode interfaces [16]. Owing to the larger surface area and structure of $rGO/TiO_2$ nanocomposites, it could be beneficial to achieve a higher specific capacitance.

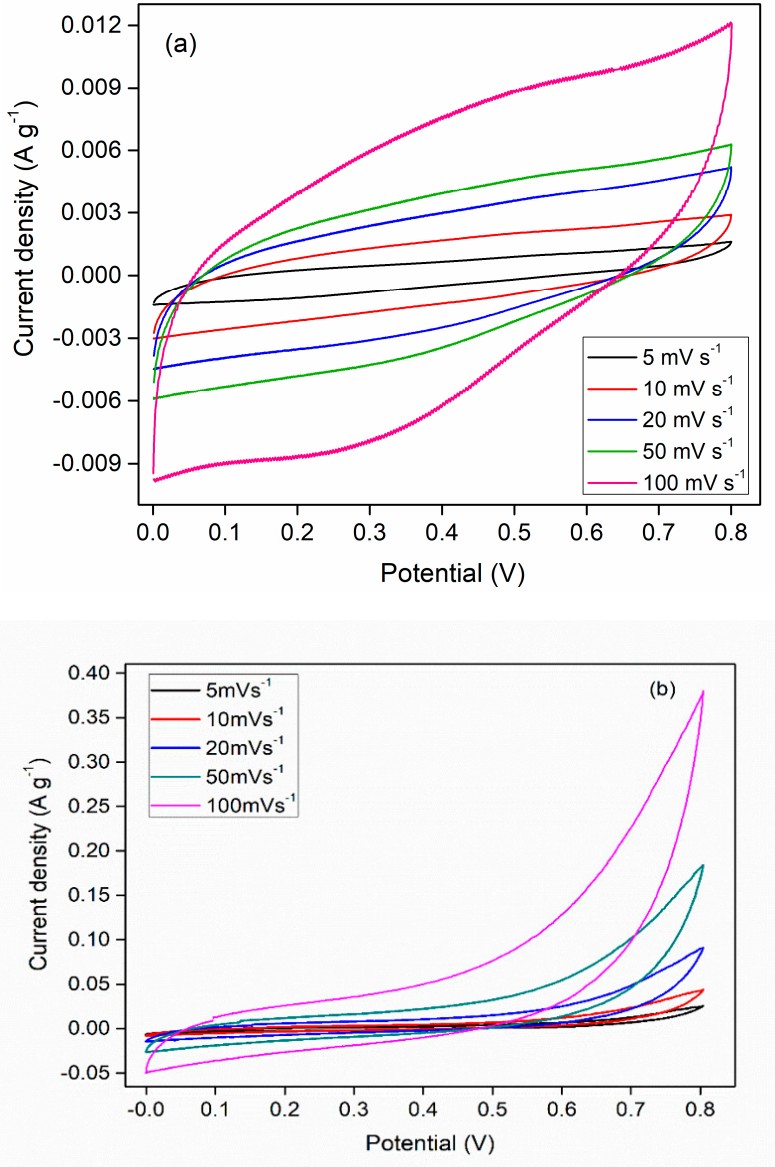

**Figure 4.** CV profile of (**a**) $TiO_2$ and (**b**) $rGO/TiO_2$.

The effect of the scan rate on the capacitance of the electrode materials is represented in Figure 5. As the scan rate increases, the values of the capacitance decrease [29]. At lower scanning rates, the maximum electrode surface is utilized for electrochemical reaction, whereas at higher scanning rates, a limited electrode surface is utilized for the electrochemical reaction [30]. Additionally, at higher scanning rates, the diffusion resistance is believed to increase because of meager migration and ion diffusion. Then, it could control the ions moving between the electrolyte and electrodes. Due to this fact, it was understood that at a high scan rate, the capacitance of the electrode could be decreased. The capacitance of the electrode was determined using equation included in reference [31].

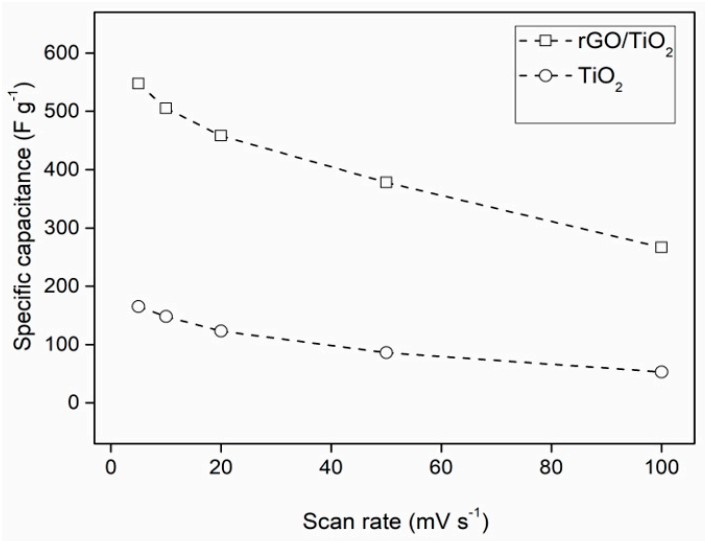

**Figure 5.** The relationship between scan rate and specific capacitance.

The profile of the scan rate and the capacitance of the electrodes is depicted in Figure 5. For $TiO_2$ based electrodes, the specific capacitances are found to be 165, 148, 123, 86, and 53 F $g^{-1}$ at 5, 10, 20, 50, and 100 mV $s^{-1}$, respectively; for $rGO/TiO_2$ based electrodes, the specific capacitances are calculated as 547, 505, 457, 378 and 267 F $g^{-1}$ at 5, 10, 20, 50 and 100 mV $s^{-1}$, respectively. Based on the above-calculated values, it is learned that the specific capacitance of the $rGO/TiO_2$ based electrode can achieve a higher value when compared to the $TiO_2$ electrode. The increased capacitance of $rGO/TiO_2$ is due to the wide surface area of the nanocomposite, facilitating the easy migration of electrolyte ions into the electrode interface. Furthermore, faradaic behavior due to the redox reaction of $TiO_2$ in the nanocomposite could enhance the specific capacitance. Furthermore, rGO in the nanocomposite could shorten the conducting path for electron transfer within the electrodes.

## 4. Galvanostatic Charge-Discharge (GCD)

The experimental of GCD is accomplished to assess the charge and discharge characteristics of $TiO_2$ and $rGO/TiO_2$ electrode materials. This experiment is implemented at a current density ranging from 1 mA $g^{-1}$ to 7 mA $g^{-1}$. It can be seen that the profiles in Figure 6a seem to be nonlinear [32] and they do not appear in triangular shapes, which are required to ascertain the EDLC type [33,34]. The nonlinear shapes of the GCD results are mainly due to the reversibility of faradaic redox reaction [35,36]. At all the current densities, the potential of the electrodes rises during charging and falls during discharging. The capacitance of electrodes was found from discharge time as per the equation included in reference [37].

The GCD profiles of the $rGO/TiO_2$ sample at diverse current densities are presented in Figure 6b. GCD curves of the $rGO/TiO_2$ sample seem to be in an almost triangular profile, which proves that the capacitance is contributed by EDLC as well as pseudo capacitance. The iR drop is not found noticeably in all GCD profiles and it indicates that $rGO/TiO_2$ electrode has good electrical conductivity and very low internal resistance. Additionally, the addition of rGO into $TiO_2$ provided a greater surface area for ion diffusion and improved the electrical properties of $TiO_2$. In GCD profiles, it is divulged that the discharge times in $rGO/TiO_2$ electrodes at different current densities are lengthier than those in $TiO_2$ electrodes. The longer discharge times could be advantageous for achieving a higher specific capacitance. These enhanced electrochemical characteristics of $rGO/TiO_2$ electrodes are attributed to the larger surface of active material, faster ion transfer, and improved intercalation of $TiO_2$ between the rGO sheets. In addition, it is necessary to determine the best proportion of rGO which can be combined with $TiO_2$, to achieve a

maximum value of specific capacitance. Higher loading of rGO with $TiO_2$ could cause agglomeration, resulting in reduced surface area of electrode material. In the case of a lower concentration of rGO, it could offer only a confined surface area for the charge transfer. The profile of the capacitance and current density for the three electrodes is illustrated in Figure 7. The capacitances of the $TiO_2$ sample were calculated as 165, 135, 94, 62, and 27 F $g^{-1}$ at 1, 2, 4, 5 and 7 mA $g^{-1}$, respectively; for the rGO/$TiO_2$ sample, the specific capacitances were found to be 531, 502, 434, 377 and 237 F $g^{-1}$ at 1, 2, 4, 5 and 7 mA $g^{-1}$, respectively.

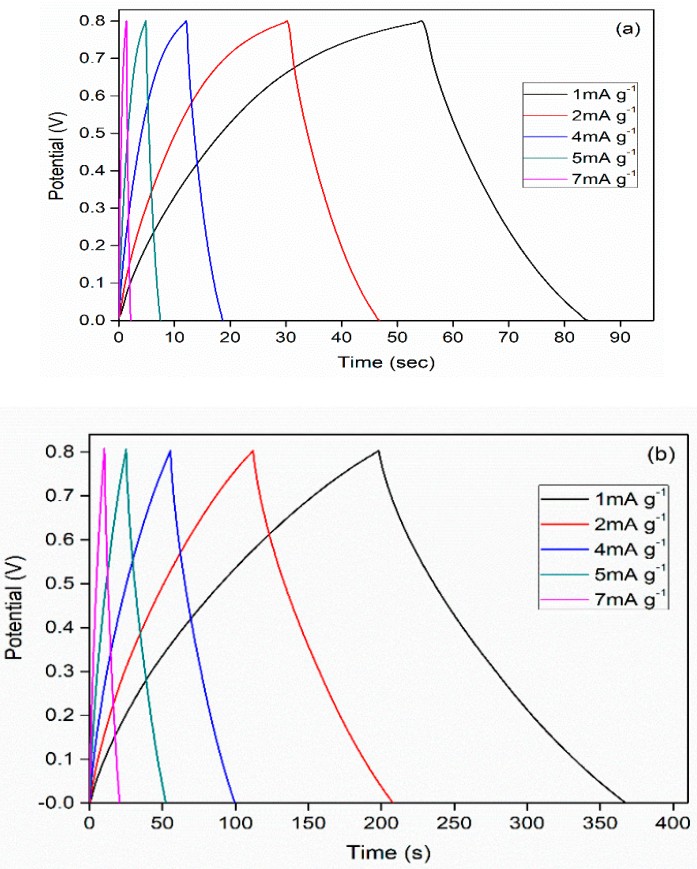

**Figure 6.** (**a**) Galvanostatic Charge-discharge curve of $TiO_2$ (**b**) Galvanostatic Charge-discharge curve of rGO/$TiO_2$.

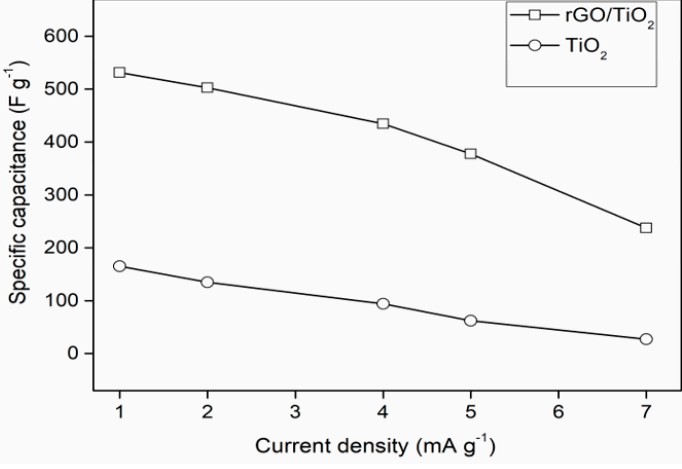

**Figure 7.** Current density versus specific capacitance.

From Figure 7, it is mentioned that as the current density increases the capacitance decreases and these reduced values of the specific capacitance concern the ion-exchange mechanism [38,39]. At a low current density, the electrolyte ions are given adequate time to access the active surface of the electrode materials completely. At a high current density, the electrolyte ions are unable to diffuse entirely into electrode materials owing to insufficient time for the charge transfer. The fall in capacitance of the $TiO_2$ electrode at 2, 4, 5, 7 mA $g^{-1}$ are calculated as 81.79%, 56.97%, 37.58%, and 16.35%, respectively, when compared to its initial capacitance value obtained at 1mA $g^{-1}$. Likewise, for rGO/$TiO_2$ the fall in capacitance at 2, 4, 5, and 7mA $g^{-1}$ are evaluated as 94.52%, 81.73%, 70.98% and 44.63%, respectively, compared with its initial specific capacitance value attained at 1mA $g^{-1}$. The capacitance of the rGO/$TiO_2$ electrode proved a superior performance compared with $TiO_2$. The presence of rGO in the nanocomposite (rGO/$TiO_2$) could increase the conductivity. In addition, rGO provides a larger surface area along its length and as a result, it reduces the pathway distance to the ions migrating between electrolyte and electrode material. Thus, it can be stated that the enhanced capacitance of the electrodes relies on the size, structure, and conductivity of electrode materials.

## 5. Electrochemical Impedance Spectroscopy (EIS)

The charge transport kinetics of electrodes was evaluated by means of EIS analyses. EIS measurements were taken with the frequency range between 0.01 and 100 kHz at the electrode potential in the $Na_2SO_4$ electrolyte solution. Nyquist plots of the EIS study provide information regarding Equivalent Series Resistance (ESR), the interaction between electrolyte-electrode, and interfacial effects. The presence of semicircles in the Nyquist plots (seen in Figure 8) in the high-frequency range indicates an interfacial ion migration process and a straight line at the low-frequency range indicates the diffusion process of electrolyte ions into the electrode inner parts.

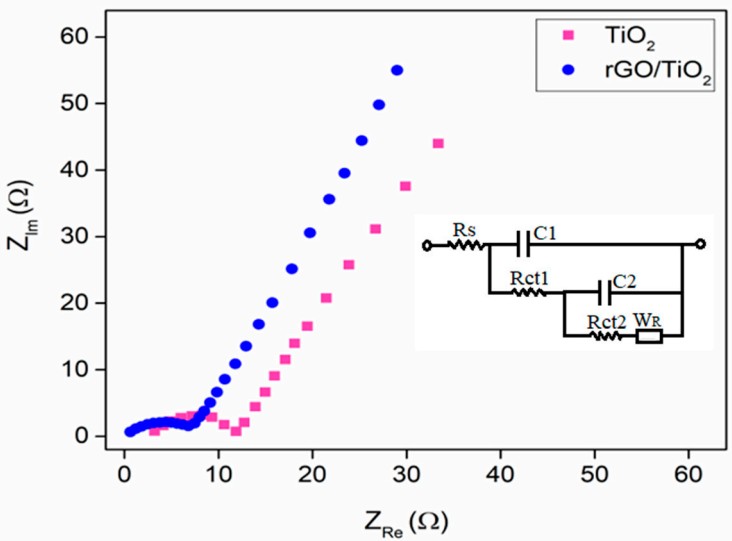

**Figure 8.** Nyquist plot of $TiO_2$ and rGO/$TiO_2$.

In Figure 8, the rGO/$TiO_2$ electrode shows a smaller semicircle compared to the $TiO_2$ electrode showing a larger charge transfer capacity and very low internal resistance. In addition, a straight line for the rGO/$TiO_2$ electrode in the low-frequency region approaches the imaginary axis, revealing the ideal capacitive behavior with low diffusion resistance when compared to $TiO_2$ electrodes. In the equivalent circuit of the rGO/$TiO_2$ electrode (shown in the inset of Figure 8), Rs exhibits solution resistance; Cl and $C_2$ represent double-layer capacitance, and Rct1 and Rct2 represent the charge transfer resistance of the coating, respectively. The $W_R$ is the Warburg element that was associated with the diffusion of ions through the diffusion layer. The ESR values of rGO/$TiO_2$ and $TiO_2$ electrodes were

observed as 0.64 $\Omega$ and 3.2 $\Omega$, correspondingly. The Rct values of rGO/TiO$_2$ and TiO$_2$ were estimated as 6.7 $\Omega$ and 8.3 $\Omega$, correspondingly. A lower ESR of the rGO/TiO$_2$ electrode divulges the enhanced electrochemical performance and good operational rate for energy storage applications. The low resistance of the rGO/TiO$_2$ electrode has shown enhanced electrical conductivity when compared to the TiO$_2$ electrode. It is mainly accredited to the electrical conductivity of rGO and its morphology. Due to this fact, it could be beneficial in improving the fast ion transport and charge transfer.

## 6. Cycle Stability

It is inevitable to probe the charge-discharge characteristics of the electrodes being used in supercapacitors for long-run operations. The charge-discharge tests on TiO$_2$ and rGO/TiO$_2$ electrodes were carried out by running 5000 cycles continuously, at 1 mA g$^{-1}$. Figure 9 shows the variation in capacitance and the number of cycles. The capacitance values of the electrode were calculated every 1000 cycles. The specific capacitance values of TiO$_2$ electrodes for 1000, 2000, 3000, 4000, and 5000 cycles were found to be 153, 146, 135, 128, and 117 F g$^{-1}$, respectively. The capacitance retentions of the TiO$_2$ electrode after 1000, 2000, 3000, 4000 and 5000 cycles were 93.29%, 89.02%, 82.31%, 78.05% and 71.34% of the initial value, respectively. The specific capacitance values of the rGO/TiO$_2$ electrode for 1000, 2000, 3000, 4000 and 5000 cycles were observed as 528, 522, 515, 509 and 499 F g$^{-1}$, respectively. The capacitance retentions of rGO/TiO$_2$ electrode after 1000, 2000, 3000, 4000 and 5000 cycles were 99.43%, 98.31%, 96.98%, 95.85% and 93.97% of the initial value, respectively.

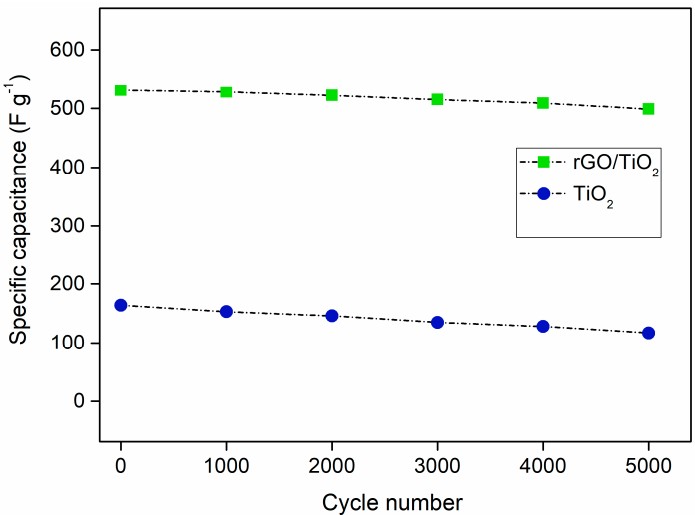

**Figure 9.** Specific capacitance versus cycle number.

Here, it is reported that the capacitance values are reduced when charge-discharge cyclic operations are increased. From the past literature, it is observed that charging and discharging could cause particle growth resulting from mass transfer and recrystallization [40]. Therefore, the active surface area of the electrode is reduced. Further, the capacitance decay during long-run operations might be attributed to chemical dissolution and intercalation/deintercalation induced material pulverization. After 5000 cycles, capacitance retention rates of 71% and 94% were achieved for TiO$_2$ and rGO/TiO$_2$ electrodes, respectively. It is palpable that the rGO/TiO$_2$ electrode has shown a better stable performance after 5000 cyclic operations when compared to TiO$_2$. The improved cycle stability of the rGO/TiO$_2$ electrode is accredited to porous structure and morphology, which provide a greater surface-mass ratio. Thereby, it could promote the electrochemical reaction for the complete electrode material. Further, it is inferred that a slight reduction in the capacitance was credited to the exfoliation of the active materials. Accordingly, it is inferred

that the choice of active material plays a prime role in determining the life cycle of the supercapacitor.

## 7. Ragone Plot

The power density versus energy density for $TiO_2$, and $rGO/TiO_2$ electrodes is represented in the Ragone plot (as shown in Figure 10). As per equations included in reference [7], energy densities and power densities were evaluated through the results received from galvanostatic tests [41].

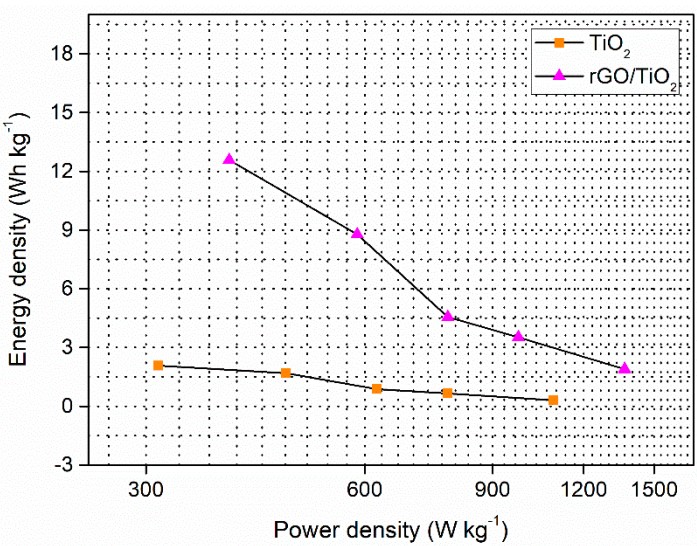

**Figure 10.** Ragone plot of the electrode.

The energy and power densities of $TiO_2$ electrode at 1, 2, 4, 5, 7 mA g$^{-1}$ were found to be 2.08, 1.69, 0.86, 0.65, 0.31 Wh kg$^{-1}$ and 311.69, 467.53, 623.37, 779.22, 1090.91 W kg$^{-1}$, respectively; the energy and power densities of $rGO/TiO_2$ electrode at 1, 2, 4, 5, 7 mA g$^{-1}$ were estimated as 12.58, 8.78, 4.55, 3.52, 1.89 Wh kg$^{-1}$ and 390.47, 585.71, 780.95, 976.19, 1366.66 W kg$^{-1}$, respectively. Based on the results, it is divulged that the specific energy of the $rGO/TiO_2$ electrode is higher than that of the $TiO_2$ electrode. At low current densities, high energy density was gained and it could be ascribed to the better intercalation/deintercalation of ions, moving between electrolyte-electrode. At high current density, low energy density was attained [30,42] and it could be attributed to increased resistance (Rct) and the limited use of the electrode. A larger energy density in the $rGO/TiO_2$ electrode resulted from enhanced specific capacitance since it is directly proportional to the capacitance of the electrode. The higher and lower energy densities of the $rGO/TiO_2$ electrode range from 12.58 to 1.89 Wh kg$^{-1}$, respectively. Owing to this reason, $rGO/TiO_2$ electrodes can be recommended for use in commercial supercapacitors.

## 8. Conclusions

To enhance the capacitive behaviors of the $TiO_2$ electrode material, rGO was combined with it, as rGO has high conductivity and affords a larger surface area. From the CV test, the capacitance of $rGO/TiO_2$ was observed as 547 F g$^{-1}$ at 5 mV s$^{-1}$. When compared to $TiO_2$, the $rGO/TiO_2$ nanocomposites exhibited improved capacitive performance. Similarly, the stability test divulges that after 5000 cycles, capacitance retention rates of 94% were achieved for the $rGO/TiO_2$ electrode. The GCD tests revealed that the newly as-prepared electrode material achieved a symmetric profile in charge-discharge and better cycle stability. It was reported that a higher proportion of rGO in the composite could cause agglomeration, resulting in a decreased surface area of the electrode material. In the case of a lower proportion of rGO, it could offer only a limited surface area for ion transfer. The energy density of $rGO/TiO_2$ at 1 mA g$^{-1}$ was evaluated as 12.58 Wh kg$^{-1}$.

Thus, it was concluded that TiO$_2$ can achieve higher specific capacitance, the energy density provided was highly conductive, and better electrochemical characteristics of the materials are infused with them.

**Author Contributions:** Data curation, P.A. and S.H.; Formal analysis, V.J.S.K.; Funding acquisition, W.-C.L.; Methodology, A.E.D.M.; Resources, V.J.S.K. and W.-C.L.; Software, P.A. and S.H.; Supervision, W.-C.L. All authors have read and agreed to the published version of the manuscript.

**Funding:** This research received no external funding.

**Conflicts of Interest:** The authors declare no conflict of interest.

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
