# Peer review of "The Enhanced Energy Density of rGO/TiO2 Based Nanocomposite as Electrode Material for Supercapacitor"

_electronics, doi:10.3390/electronics11111792_

Round 1

Reviewer 1 Report

Engineering a novel materials for the supercapacitor is a challenging and also highly demanding as World is moving towards green energy solution. In this context, RGO/TiO2 composite prepaeration by Anadhi et al and exploring their charge-storage performance is very timely. Prior to consider the article, authors need to address the following concerns

  1. Introduction part is very general, very lengthy and well-known to everyone in this field. Authors should concentrate on the challenges of TiO2 and TiO2/carbon composites in the existing lietratures and how it has solved in this work.
  2. From XRD, authors should explain what happens to the brookite phase in the composite with respect to the TiO2 nanoparticles. Need more explanation.
  3. Provide the normalized mass loading in the unit of mg/cm2
  4. Is there any possible explanation on the CV shape changes in the composite compared to the nanoparticles. particularly, in the zone of 0.5 to 0.8 V of nanocomposite.
  5. taking about the energy density from an electrode measured in three-electrode system is not meaningful at all. It is advised to make a device and calculate the gravimetric capacitance, cycle life, energy density and power density. 
  6. it is also recommended to provide some explanation on the metal oxide formation on the carbon surface. Please follow and cite the reference 10.1016/j.jiec.2018.12.008
  7. provide the information on the Rct and ESR calculation. Are they fitted with an equivalent circuit modelling?
  8. there are problem in spacing and comma throughout the manuscript

Author Response

Thanks for your valuable comments/ suggestions. Authors' answers as following:

  1. Some portion was deleted and the new lines regarding TiO2 were included in the revised manuscript. (Page: 3, Line: 130-135)
  2. Mentioned in the revised manuscript. (Page: 8, Line:282-284)
  3. Mentioned. (Page: 5, Line: 218)
  4. Explained. (Page: 8, Line: 311-315)
  5. Thanks for your suggestion. In my future work, I shall certainly make a device to study the energy storage capacities. Two new references [38, 42] were cited. (Page: 5, Line: 220)
  6. Cited the reference [39]. (Page: 4, Line: 195)
  7. Yes, it is fitted. The equivalent circuit is inserted in Figure 8. (Page: 13,Line: 423-427)
  8. The manuscript was completely verified and corrected.

Reviewer 2 Report

The manuscript entitled "The Enhanced Energy Density of rGO/TiO2 Based Nanocompo- 2 site as Electrode Material for Supercapacitor". Dear Editor and authors, the work is well written and with very good characterization and electrochemical data. I see a very good paper which can attract large attention from the people interested in electrochemistry and materials applied in supercapacitors. However, before I recommend its acceptance, some points must be clarified and a moderate revision is needed.

Some other issues that need to be addressed are:

  1. The main problem statement and justification for the research has not been clearly stated.
  2. It is not clear the contribution of the manuscript to the empirical literature.
  3. Would you explicitly specify the novelty of your work? The main novelty in this work must be clearly pointed out in the introduction.
  4. The authors should mention on the concept of this work with the progress against the most recent state-of-the-art similar studies.
  5. I find there is no convincing link between the motivations for doing the paper and the way it has been conducted as well as the conclusions reached.
  6. The limitation of this study needs to be provided as well.
  7. Why three electrode system was used? The authors know that this boosts a lot the supercapacitors’ performances. Also, this is not realistic at all. Give a good explanation for that.
  8. The citation of only 35 references in this good paper I find not sufficient. At least 40 should be cited. Based on that the 2 references below can help the authors to improve the discussion and the quality of the manuscript.

https://doi.org/10.3390/electronics8020254

https://doi.org/10.3390/nano11020424

Author Response

Thanks for your valuable comments/ suggestions. Authors' answers as following:

  1. Rewritten in the revised manuscript. rGO was combined with TiO2 to improve the electrochemical performance. (Page: 3, Line: 140-152)
  2.  In the introduction section, energy storage devices were discussed, and then, the evolution of supercapacitors was discussed. Metal oxides-based supercapacitors and the drawback of TiO2. In the end, the novelty of the work was mentioned. To improve the performance of titanate, rGO was incorporated so as to enhance the performance. Some new lines regarding TiO2 were included in the revised manuscript. (Page: 3, Line: 130-135)
  3. It was stated in the revised manuscript. (Page: 3, Line: 143-146)
  4. Here, rGO was combined with TiO2 and it was done by wet chemical method. To the best of our knowledge, this method was not used before to fabricate rGO/TiO2 nanocomposite for supercapacitor application. Also, this method is simple, fast, and cost-effective for large-scale industrial production. (Page: 3, Line: 143-152)
  5. First, it was focused to increase the surface area of the active sites of TiO2. To achieve the above, rGO was chosen as a supporting material in order to increase the conductivity and provide a larger surface area for active sites during ion transfer.Based on the results, it was found that the specific capacitance of the nanocomposite was improved and energy density was also improved. 
  6. Here, in the present study, only one mass loading of rGO was considered and in the future, it is aimed to vary the mass of rGO into TiO2. Then, it will be found the optimum mass loading of rGO for the better electrochemical performance of the electrodes of the supercapacitor. 
  7. Thanks for your suggestion. In my future work, I shall certainly make a device to study the energy storage capacities. Two new references [38, 42] were cited. (Page: 5, Line: 220)

  8. Two suggested references [40, 41] were included in the revised manuscript. After revision, there are 42 references in the manuscript. (Page: 17, Line: 586-590)
